# Polygenic selection to a changing optimum under self–fertilisation

**Matthew Hartfield**[1]*, **Sylvain Glémin**[2,3]

**1** Institute of Ecology and Evolution, The University of Edinburgh, Edinburgh, United Kingdom, **2** Université de Rennes, Centre National de la Recherche Scientifique (CNRS), ECOBIO (Ecosystèmes, Biodiversité, Evolution) – Unité Mixte de Recherche (UMR) 6553, Rennes, France, **3** Department of Ecology and Evolution, Evolutionary Biology Center, Uppsala University, Uppsala, Sweden

* m.hartfield@ed.ac.uk

**Data Availability Statement:** Simulation code and data processing scripts are available from https://github.com/MattHartfield/Polygenic-Selfing.

**Funding:** MH is supported by a NERC Independent Research Fellowship (NE/R015686/1) and a UKRI Frontier Research Guarantee Grant (EP/X027570/

## Abstract

Many traits are polygenic, affected by multiple genetic variants throughout the genome. Selection acting on these traits involves co–ordinated allele–frequency changes at these underlying variants, and this process has been extensively studied in random–mating populations. Yet many species self–fertilise to some degree, which incurs changes to genetic diversity, recombination and genome segregation. These factors cumulatively influence how polygenic selection is realised in nature. Here, we use analytical modelling and stochastic simulations to investigate to what extent self–fertilisation affects polygenic adaptation to a new environment. Our analytical solutions show that while selfing can increase adaptation to an optimum, it incurs linkage disequilibrium that can slow down the initial spread of favoured mutations due to selection interference, and favours the fixation of alleles with opposing trait effects. Simulations show that while selection interference is present, high levels of selfing (at least 90%) aids adaptation to a new optimum, showing a higher long–term fitness. If mutations are pleiotropic then only a few major–effect variants fix along with many neutral hitchhikers, with a transient increase in linkage disequilibrium. These results show potential advantages to self–fertilisation when adapting to a new environment, and how the mating system affects the genetic composition of polygenic selection.

## Author summary

Many biological traits of scientific interest are polygenic, which are influenced by multiple genetic variants present throughout the genome. Emerging whole-genome data from several species is shedding light on how such traits respond to selection, traditionally through co-ordinated changes in variant frequencies. However, many species in nature reproduce via self-fertilisation, where hermaphrodite individuals produce both male and female gametes that can be used to propagate without mates. This reproductive mode can reduce population-level diversity and the reassorting effects of recombination, which affects how polygenic traits respond to selection. In this paper, we theoretically explore how polygenic selection is realised under self-fertilisation, following a shift in the environment. We first show analytically how the mating–system affects the dynamics of polygenic selection,

1). SG is supported by the CNRS. The funders had no role in study design, data collection and analysis, decision to publish, or preparation of the manuscript.

**Competing interests:** The authors have declared that no competing interests exist.

showing that there are two competing effects. First, it can expose mutations to selection more quickly, strengthening adaptation to a changing environment. Conversely, it can reduce the efficacy of selection through weakening the efficacy of recombination. We then use multi–locus stochastic simulations to investigate outcomes under more realistic scenarios, and find that high selfing can lead to higher fitness in the long–term, in contrast to classic expectations. We also investigate how many traits each variant influences, a property known as pleiotropy. If pleiotropy is absent we see that under very high levels of self-fertilisation, populations fix mutations with opposite effects on a trait. If pleiotropy is present then we instead see only a few major-effect genetic variants fixing in the population, alongside many neutral mutations. These findings provide insights into how natural populations adapt to changing environments.

## Introduction

Detecting where selection acts in the genome, along with the mechanisms at play, is a major research goal in evolutionary genetics. Many traits that affect fitness are complex and polygenic, meaning they are affected by multiple variants present genome–wide [1]. Selection acting on these traits will incur allele–frequency changes acting on these variants simultaneously. Recent years have seen major interest in polygenic selection research, aided by the appearance of large genome datasets that can identify many variants with small effect sizes, and new methods developed specifically for detecting it (reviewed by [2–6]). While theory for predicting the short term response to selection is well-established and efficient under most conditions, understanding the long-term selection response remains much more complicated and challenging, in particular because it requires knowledge of how genetic variances evolves, along with the fine–scale genetic details of a trait [7]. As a consequence, it is also difficult to link phenotype changes under selection with changes in the underlying genomic architecture. Recent progress has been made through extensive multi–locus simulations [8, 9] or new analytical developments [10–15]. In particular, Hayward and Sella [13] obtained approximations that jointly characterised the phenotypic changes through time and the allele dynamics at loci contributing to trait variance.

Most results have been obtained for random-mating populations, yet many species are hermaphroditic and capable of self-fertilisation. While commonly associated with plants where up to 50% of species self–fertilise to some degree [16], self-fertilisation is also present in animals [17], fungi [18] and algae [19]. An increasing number of genomic studies are performed on self–fertilising organisms where polygenic selection has been detected, including the highly selfing plant *Arabidopsis thaliana* [20–24] or in *Mimulus guttatus*, a mixed–mating species [25]. There have also been examples of multiple beneficial mutations fixing in linkage in highly selfing organisms. These include the localised presence of genes affecting multiple traits (including climate adaptation, root growth and flowering time) on Chromosome 3 of *A. thaliana* [26], and linked toxin-antidote pairs in self–fertilising *Caenorhabditis* nematodes [27, 28]. Given that most existing theory and inference methods assume randomly–mating populations, it may lead to inaccurate selection predictions and interpretations of genome data when applied to selfing species.

Self–fertilisation incurs several genetic effects [29, 30]; most notably, the resulting inbreeding increases homozygosity and hence also the effects of genetic drift, which manifests itself as a reduction in $N_e$ thus weakening the efficacy of selection [31–34]. The resulting homozygosity also reduces effective recombination rates and, hence, the creation of new genetic

combinations [35, 36]. These factors can affect how such species adapt to new environments [37]. [38] argued that the various genetic changes incurred by self–fertilisation will limit species' adaptive potential. In particular, selection acting on favourable mutations can be hampered through reduced diversity and increased genetic interference caused by decreased recombination [39–42]. Conversely, due to reduced recombination and quicker fixation time of favourable alleles, selection footprints can be more pronounced in selfers as they are present across longer genetic distances [43, 44]. Previous theoretical studies have tended to focus on either one– or two–locus models for mathematical tractability (but multi–locus simulation studies exist, e.g., [40]). However, the effect on positive selection acting over the whole genome is currently unclear. Under extremely high self–fertilisation, recombination can be so rare so as to effectively cause selection to act on the entire genome as a single linked unit. For example, selective sweeps affecting entire chromosomes have been detected in the selfing nematode *Caenorhabditis elegans* [45]. This process is related to the 'genotype selection' model of [46]; when recombination is sufficiently weak and epistasis exists between mutations, a subset of genotypes have the highest fitness and proliferate in the population. In this scenario, reduced recombination can offer individuals a competitive advantage.

These genetic effects of selfing complicates theoretical developments compared to randomly-mating populations. In particular, genetic associations generated across the genome violate several assumptions made in classic models and cannot be neglected (at least under a high selfing rate). Typically, a partial selfing population is composed of classes of individuals with different inbreeding levels, which generates correlations in homozygosity across loci (also called 'identity disequilibrium'; [47]). On the one hand, this increase in homozygosity helps purge deleterious variation; on the other hand, reduced effective recombination due to selfing amplifies the "Bulmer effect", the reduction in genetic variance due to negative linkage disequilibrium [48, 49]. As a consequence, selfing reduces the equilibrium genetic variance maintained for quantitative traits under purifying selection [50, 51]. Partial selfing also violate some assumptions of the Breeder's equation by making parent-offspring regression non–linear, thus complicating predictions of the selection response [48, 49].

Hence, polygenic responses to selection have thus been mainly explored through simulations. [52] found that cryptic genetic variation generated by negative associations can be re–mobilized following occasional outcrossing, aiding initial adaptation to a new environment. Self–fertilisation also aids a population establishing in a new location following a bottleneck, mainly through the purging of the genetic load [53]. These two studies focused on the phenotypic response to selection but did not investigate the underlying genomic dynamics. In contrast, [54] investigated how self–fertilisation affects the clustering of polygenic variants underpinning local adaptation within a genome, but in the context of populations at migration-selection equilibrium. Given these disparate studies, there is a need for a general theoretical investigation into the dynamics of polygenic selection under self–fertilisation, especially making the link between the phenotypic response and the underlying allelic dynamics, which can be used to interpret the genetic signatures of polygenic selection from next–generation sequence data. In addition, these aforementioned studies only considered selection on a single trait, while adaptation to a new environment likely involve several pleiotropic traits that are simultaneously affected by mutation.

Here, we investigate the general behaviour of polygenic adaptation to a new environment under self–fertilisation. Specifically, we investigate whether increased to what extent self–fertilisation helps or hinders polygenic adaptation, and its resulting influence on the underlying genetic composition. We begin by developing simple heuristic one-locus and two-locus models to help understand the selection response. However, a full analytical treatment of the

problem is out of reach, so we rely on extensive simulations following both phenotypic evolution and the underlying genomic dynamics.

## Analytical results

### Two-locus model

We start by presenting a two–locus analysis of quantitative traits under stabilising selection and self–fertilisation. Although this system is highly simplified, it is commonly used as a stating point for quantitative genetics models as it yields insights into how associations between loci (here caused by either low recombination and/or high self–fertilisation) changes the polygenic selection response; [7]). We consider two biallelic loci in a diploid individual carrying alleles denoted $a$, $A$ and $b$, $B$ respectively. Alleles $a$, $b$ are wildtypes, while $A$, $B$ carry mutations that contribute $\alpha_A$, $\alpha_B$ to a quantitative trait. We denote the relative frequencies of the four possible haplotypes $ab$, $Ab$, $aB$ and $AB$ by $x_1 \ldots x_4$ respectively; we further denote $g_{ij}$ as the relative frequency of the genotype consisting of haplotypes $i$, $j$. We can convert haplotypes to allele frequencies using standard equations that also consider the degree of linkage disequilibrium between alleles [55]; for example, $x_4 = p_A p_B + \delta$ for $p_A$, $p_B$ the frequencies of alleles $A$, $B$, and $\delta$ the level of linkage disequilibrium. We convert genotypes frequencies to allele frequencies using the approximation that inbreeding increases the probability of identity–by–descent of two haplotypes by a factor $F$:

$$
\begin{aligned}
g_{ii} &= x_i^2 + F x_i (1 - x_i) && \text{for } i \in \{1, \ldots, 4\} \\
g_{ij} &= 2 x_i x_j (1 - F) && \text{for } i \neq j
\end{aligned}
\tag{1}
$$

Although this approximation does not take into account recombination during self–fertilisation, it is nevertheless accurate unless self–fertilisation and recombination are both high [56–58].

We then use the recursion formulas in [59] to determine how each genotype changes every generation due to selection and reproduction; we will add on mutation afterwards. Phenotypes of genotype $ij$ are given by the sum of their individual allele effects, minus the optimum value $z_0$. For example, the phenotype of $g_{44}$ (i.e., $AB/AB$) is $2\alpha_A + 2\alpha_B - z_0$. Note that trait values were not standardised so that homozygote genotypes had the same magnitude of effect but in opposite directions, as is often used in quantitative genetics models [1]. Fitness is determined by stabilising selection:

$$
w(z) = exp(-z^2 / (2V_s))
\tag{2}
$$

for $z$ a phenotype value and $V_s$ the strength of stabilising selection. Without loss of generality, we re-scale the effect size of mutations and the distance to the optimum by $\sqrt{V_s}$, so $\gamma_A = \alpha_A / \sqrt{V_s}$, $\gamma_B = \alpha_B / \sqrt{V_s}$, and $d = z / \sqrt{V_s}$. To obtain tractable solutions, we assume weak selection (i.e., $\gamma_A, \gamma_B = \mathcal{O}(\zeta)$). We then use the aforementioned transformations (Eq 1) to then find how allele frequencies and linkage disequilibrium change every generation. The full list of recursions for the two–locus model is presented in S1 File.

At the optimum ($d = 0$), all mutants are deleterious so we can also assume low allele frequencies and linkage disequilibrium (technically, $p_A, p_B = \mathcal{O}(\zeta) \ll 1$ and $\delta = \mathcal{O}(\zeta^2)$). We first calculate the change in linkage disequilibrium over one generation, $\Delta\delta$, which is given as:

$$
\Delta\delta = -(1 + 3F) p_A p_B \gamma_A \gamma_B - \frac{\delta}{2} \Gamma + \mathcal{O}(\zeta^3)
\tag{3}
$$

where $\Gamma = (1 + 3F)(\gamma_A + \gamma_B)^2 + (1 - F)r(2 - (\gamma_A + \gamma_B)^2)$. Eq 3 demonstrates that in the absence

of initial linkage disequilibrium (hereafter LD), selection leads to negative LD if the two alleles are aligned in the same direction (i.e., $\gamma_A \gamma_B > 0$) and positive LD if they are in opposite directions (i.e., $\gamma_A \gamma_B < 0$). These results are similar to the 'Bulmer effect' [60, 61] and amplified by self–fertilisation. The second term shows that existing LD is further broken down by both selection and recombination. Self–fertilisation has opposing influences on these factors, as denoted by Γ: Bulmer effects are amplified by selfing but their breakdown by recombination is diminished by it. To determine how this balance will play out, we can look at the rate of change of Γ as a function of $F$:

$$\frac{\partial \Gamma}{\partial F} = 3(\gamma_A + \gamma_B)^2 - r(2 - (\gamma_A + \gamma_B)^2) \tag{4}$$

A bit of algebra shows that Eq 4 is positive if $r < 3(\gamma_A + \gamma_B)^2/(2 - (\gamma_A + \gamma_B)^2)$. Hence, if recombination is sufficiently weak then self–fertilisation will further enhance negative LD between loci, otherwise it will inhibit it. The latter case corresponds to the conditions of quasi–linkage equilibrium, where $\gamma_A, \gamma_B \ll r(1 - F)$. Denoting this value $\delta_{QLE}$, it is given by:

$$\delta_{QLE} \approx -\frac{(1 + 3F)\gamma_A \gamma_B}{(1 - F)r} p_A p_B \tag{5}$$

Eq 5 is approximate but it clearly shows how increased selfing greatly magnifies the extent of LD caused by stabilising selection between two loci. A similar but more accurate expression was obtained by [51]. LD is low over two loci but, over a larger genome, the sum of these pair-wise interactions will greatly increase genome–wide LD [51]. We can also derive the allele frequency changes in a similar manner, but these terms are quite lengthy and involve the main parameters in a complicated manner (S1 File).

Now, consider the case where the optimum has shifted. Without loss of generality, we work with the scaled version of the distance, $d_0 = z_0/\sqrt{V_s}$, and assume $d_0 > 0$. When mutational effects are small compare to the distance to the new optimum ($\gamma_A, \gamma_B \ll d_0$) the dynamics mainly corresponds to directional selection (i.e., the 'rapid phase' in [13]). The change in the frequency of $A$ can be given as:

$$\Delta p_A = d_0(1 + F)(\gamma_A p_A(1 - p_A) + \delta \gamma_B) + \mathcal{O}(\zeta^2) \tag{6}$$

with a similar term for $\Delta p_B$. Eq 6 demonstrates how, here too, self–fertilisation has contrasting effects on the spread of selected alleles following an optimum shift. First, heightened homozygosity will increase direct selection acting on an allele, as denoted by the $1 + F$ term (note that if we also include the effect on drift, i.e., considering the product $N_e \Delta p_A$ for $N_e = N/(1 + F)$ [34, 62], the two effects of selfing cancel out as for additive selection [32, 33]). Secondly, selection is altered due to LD (denoted by the $\delta \gamma_B$ term). Because LD is generally negative before the optimum shift (Eq 5), $\delta \gamma_B$ is always of the opposite sign of $\gamma_A$, slowing down the initial allele dynamics. The subsequent dynamics depends on the evolution of LD under directional selection, which is given by:

$$\Delta \delta \approx \delta(\kappa(1 + F) - (1 - F)r) \tag{7}$$

with $\kappa = d_0(\gamma_A(q_A - p_A) + \gamma_B(q_B - p_B))$ for $q_A = 1 - p_A$ (similar for $q_B$). The first term of Eq 7 (containing $1 + F$) shows that selfing amplifies the magnitude of LD depending on the sign of $\kappa$, which also depends on the imbalance between allele frequencies; if both alleles are at a frequency of 1/2 then this term is zero (see also similar results by [63, 64]). The second term indicates how selfing reduces the effect of recombination on breaking down LD.

$\kappa$ is a key parameter determining the long–term behaviour following an optimum shift. First, we focus on pairs of alleles with effects aligned in the same direction as the optimum shift ($\gamma_A, \gamma_B > 0$). Initially, they are rare and in negative LD, thus selection mainly acts on $Ab$ and $aB$ haplotypes that increase in frequency, reinforcing the initial negative LD. Selfing reinforces selective interference by speeding up the increase in frequency of both haplotypes (first term in 7) and preventing the production of the best $AB$ haplotype (second term in Eq 7). When alleles reach sufficiently high frequency ($\kappa$ approaches zero), selection then removes the initial negative LD between $A$ and $B$, with this effect reinforced by selfing, especially when mutations have strong effects. Now consider pairs of alleles with opposing effects ($\gamma_A > 0$ and $\gamma_B < 0$). While the spread of $A$ is still impeded by LD as $\delta\gamma_B < 0$, $\kappa$ is more likely to be smaller in magnitude due to the contrasting effect sizes, weakening the selective effect in impeding the spread of selected mutations.

While the effect of selfing on the whole dynamics is complex, this simple model predicts that the mating system will (i) reinforce LD between alleles with opposing effects (analogous to the 'Bulmer effect'); (ii) slow down initial allele dynamics, due to this LD reinforcement; and (iii) favour the fixation of alleles with opposing effects. However, note that these qualitative predictions may be invalid under high selfing, partly because we do not consider the full genotype dynamics (see assumptions underpinning Eq 1), and reduced linkage will violate QLE assumptions.

## Allelic dynamics in a multi-locus context

To extend this analysis, we now consider the dynamics of a single allele in a multi-locus context. This is a standard modelling approach and one recently employed by [13]; see S2 File for the full calculations.

Consider a quantitative trait affected by $n$ bi-allelic loci with additive effect, where $\alpha_i$ is the effect of allele $A_i$ in frequency $x_i$, the mean phenotype of the population is $z = \sum_i 2\alpha_i x_i$. This can be generalized to multiple traits as in [51]. Considering one focal locus, the mean phenotypic effect of the genetic background of the $n - 1$ other loci is simply $z_i = z - 2\alpha_i x_i$. In standard models, associations are neglected so the background is the same for the two alleles, so the phenotype of the three genotypes are:

$$
\begin{aligned}
a_i a_i \quad &: z - 2\alpha_i x_i \\
A_i a_i \quad &: z + \alpha_i(1 - 2x_i) \\
A_i A_i \quad &: z + 2\alpha_i(1 - x_i)
\end{aligned}
$$

However, previous studies [50–52] and the above two-locus analysis show that associations cannot be neglected, at least for a sufficiently high selfing rate. Here we propose a new approach to include associations in this framework. In the more general case we note that allele $A_i$ is associated with the background $z_i + \beta_{A,i}$, and $a_i$ with the background $z_i + \beta_{a,i}$. Because the background coefficient must satisfy $\beta_{A,i}x_i + \beta_{a,i}(1 - x_i) = 0$ we can write $\beta_{A,i} = \beta_i(1 - x_i)$ and $\beta_{a,i} = -\beta_i x_i$. The phenotype of the three genotypes can thus be written as:

$$
\begin{aligned}
a_i a_i \quad &: z - 2\alpha_i x_i - 2\beta_i x_i \\
A_i a_i \quad &: z + \alpha_i(1 - 2x_i) + \beta_i(1 - 2x_i) \\
A_i A_i \quad &: z + 2\alpha_i(1 - x_i) + 2\beta_i(1 - x_i)
\end{aligned}
$$

In the following we use the same re-scaling as above and assume weak selection, implying that both $\alpha_i, \beta_i = o(\sqrt{V_s})$. In S2 File we show that at distance $D$ from the optimum, the change

in allelic frequency at locus $i$ is given by:

$$\Delta x_i(t) = \frac{1}{V_s}(1 + F)(\alpha_i + \beta_i(t))D(t)x_i(t)(1 - x_i(t))$$
$$- \frac{1}{V_s}(1 + 3F)(\alpha_i + \beta_i(t))^2\left(1 - \frac{D(t)^2}{V_s}\right)x_i(t)(1 - x_i(t))(1/2 - x_i(t)) \tag{8}$$

This is equivalent to Equation 7 in [13] but including the effect of selfing and genetic associations. Importantly, the $\beta_i$ are not parameters but variables that depend on the rest of the model, so we explicitly note its dependence on time as with the distance to the optimum. Following [13] we can also derive the equation for the change in the distance to the optimum:

$$\Delta D(t) = -\frac{1}{V_s}\left((1 + F)V_g(t) + C_{LD}(t)\right)D(t)$$
$$+ \frac{1}{V_s}(1 + 3F)\left(1 - \frac{D(t)^2}{V_s}\right)\left(\mu_3(t) + 2v_{2,1}(t) + v_{1,2}(t)\right) \tag{9}$$

where $V_g$ is the genic variance, which is the variance assuming Hardy–Weinberg expectations and no linkage, and $C_{LD}$ the linkage covariance, and $\mu_3$, $v_{2,1}$ and $v_{1,2}$ are central moments of third order (see S2 File). Note that in Eqs 8 and 9, the dependency on time has been explicitly added to emphasise quantities that evolve through time. In particular, $V_g$ is a variable, not a parameter of the model; unfortunately, we were unable to find an equation describing its dynamics. Similarly, we could not obtain equations for $C_{LD}$ nor for $v_{1,2}$ and $v_{2,1}$.

At equilibrium ($D = 0$) and under mutation–selection balance, allelic frequencies are given by:

$$x_i(0) = \frac{uV_s}{(1 + 3F)(\alpha_i + \beta_i(0))^2} \tag{10}$$

When associations are neglected ($\beta_i = 0$), this is equivalent to Equation 19 in [51]. More generally, as $\beta_i$ is negative and amplified by selfing, Eq 10 shows that associations reduce the purging effect of selfing.

In S2 File we show that we can also obtain the equilibrium genetic variance, $V_G$:

$$V_G = \frac{4UV_s(1 + F)}{(1 + 3F)}\frac{1}{n}\sum_{i=1}^{n}\frac{\alpha_i}{(\alpha_i + \beta_i)} \tag{11}$$

where $U = \sum_{i=1}^{n} u_i$ is the total mutation rate. $V_G$ can be decomposed into its components: the genic variance $V_g$, which is the variance assuming Hardy–Weinberg expectations and no linkage, the inbreeding covariance $V_I$ reflecting deviations from Hardy–Weinberg equilibrium, and the linkage covariance $C_{LD}$:

$$V_g = \frac{4UV_s}{(1 + 3F)}\frac{1}{n}\sum_{i=1}^{n}\frac{\alpha_i^2}{(\alpha_i + \beta_i)^2} \tag{12a}$$

$$V_I = \frac{4UV_sF}{(1 + 3F)}\frac{1}{n}\sum_{i=1}^{n}\frac{\alpha_i^2}{(\alpha_i + \beta_i)^2} \tag{12b}$$

$$C_{LD} = \frac{4UV_s(1+F)}{(1+3F)} \frac{1}{n} \sum_{i=1}^{n} \frac{\alpha_i \beta_i}{(\alpha_i + \beta_i)^2} \tag{12c}$$

When associations are neglected ($\beta_i = 0$), $C_{LD} = 0$ and we retrieve the results of [51]. Taking interactions into account, we did not obtain expression for the $\beta_i$ that depend on the other parameter of the model. Before continuing, we note that $\alpha_i$ and $\beta_i$ are of opposite sign due to stabilising selection generating negative LD ([65], and see the two–locus analysis above). Because of these assumptions, Eq 12 shows that interactions increases the genic and inbreeding variance because purging is less efficient (see Eq 10) and make the covariance component more negative. Overall, reduced purging predominates and interactions increases the genetic variance (Eq 12a), as obtained with a different approach by [51] (see their Equation 37).

Selfing also affects the dynamics of adaptation. When associations are neglected, selfing only has a re-scaling effect. For rapid phase of the dynamics (*sensu* [13]), increase in homozygosity and increases in drift (both by a factor $1 + F$) cancel out and selfing has no effect, as expected under additive selection, as shown above for the two-locus model. For the equilibrium phase, selfing increases the efficacy of selection towards the new optimum (by a factor $(1 + 3F)$), which overwhelms the increase in drift and should speed up this phase. When associations are taken into account, Eqs (8) and (9) show that selfing also has an opposite effect and reduces selection efficacy because of genetic associations (term in $C_{LD}$). However, in contrast with the equilibrium state for which qualitative predictions can be made, we cannot predict which of these effects will predominate. Hence, we use simulations to predict the overall effect of selfing on the dynamics of adaptation.

## Simulation results

To more fully understand the response to polygenic selection under more realistic scenarios, we supplement these analytical results with large–genome simulations. We used the SLiM software [66] to simulate a large diploid population ($N = 5,000$) where each individual consisted of a 25Mb chromosome consisting of alternating neutral and gene regions (a similar setup was used by [54]). Mutations in genes can either be neutral, affect the quantitative traits determining fitness, or (in some simulations) unconditionally deleterious. We defined mutation and recombination rates that were realistic compared to empirical estimates ($4 \times 10^{-8}$ and $1.98 \times 10^{-7}$, respectively), but also ensured sufficient trait–variants arose per generation to allow a polygenic selection mechanism [12]. Fitness was determined by a Gaussian function, and trait mutations either affected a single trait (no pleiotropy) or five traits (pleiotropy present). There was a burn–in period of $10N = 50,000$ generations to an initial optimum value, then the optimum shifted and the population adapted to the new value. Mutation continues after the optimum shift, so results reflect adaptation from both standing variation pre-shift and *de novo* mutation afterwards. More information is available in the Methods section.

Given the large number of parameters in our simulation model, we will introduce results in a piecewise manner. Specifically, we first investigate the general behaviour of polygenic selection under different self–fertilisation levels, then investigate how deviations from this core model affect the main results.

### Basic results: Sudden optimum shift, no background selection

**Mean fitness and inbreeding depression.** Following the optimum shift, populations quickly adapt to the new optimum irrespective of the frequency of self–fertilisation (Fig 1A).

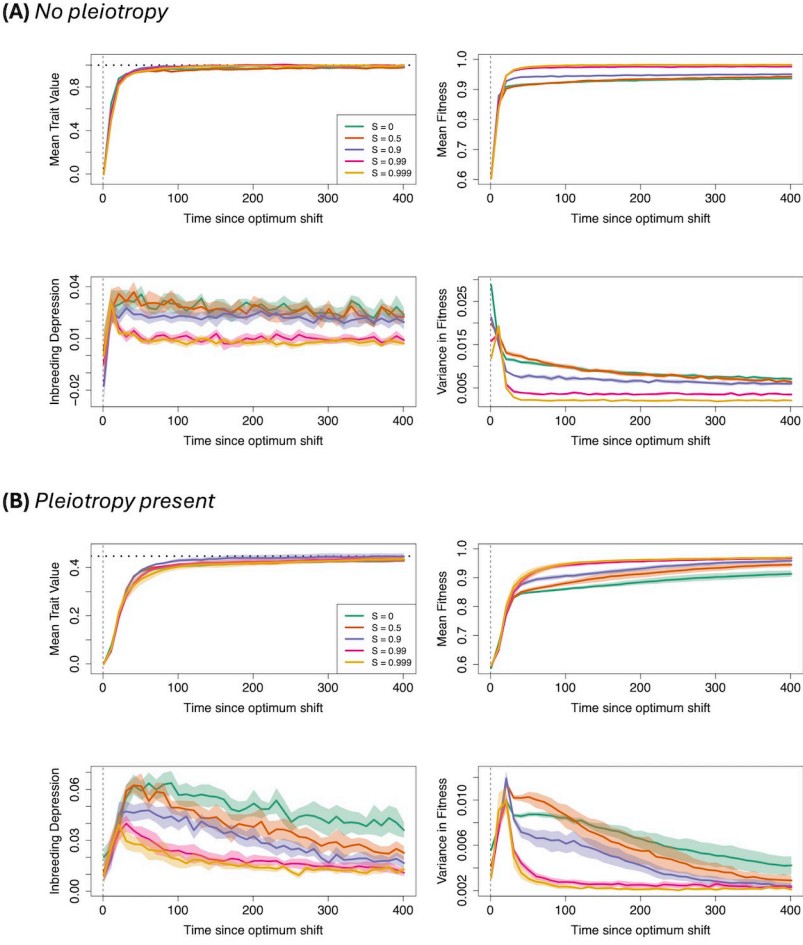

**Fig 1. Adaptation to new optimum under different self–fertilisation values.** Mean trait value (top–left; horizontal dashed lines denote new trait optima); mean fitness (top–right); inbreeding depression (bottom–left); and variance in fitness (bottom–right) after an optimum shift. Different colours represent different fractions of self–fertilisation, as denoted by the legend. Solid lines are mean values, while bands denote 95% confidence intervals. Each trait variant affected either 1 (A) or 5 (B) traits.

This corresponds to predictions for the rapid phase when genetic associations are neglected (see the explanation above for Eq 9) and more generally, the fact that selfing increases the rate of spread of selected alleles, but also increases genetic drift (Eqs 6, 8 and 9). Despite these similarities, the mean fitness is highest and inbreeding depression lowest for highly self–fertilising populations (greater than 99%), showing how self–fertilisating populations can exhibit high fitness after the optimum has been reached. One reason for this behaviour is that the variance in individual fitness is reduced under high self–fertilisation, limiting the proportion of maladaptive individuals with extreme trait values, and reducing the mutation load and inbreeding depression [50, 51]. Results are qualitatively the same when pleiotropy is present (Fig 1B), albeit the time to reach the optimum, and subsequent decays in inbreeding depression and variance in fitness, take a longer amount of time. Inbreeding depression also increases with pleiotropy, as previously observed by [67].

**Genetic variance and its decomposition.** With one trait (no pleiotropy; Fig 2A), genetic variance increases after the optimum shift and remains at elevated values for most self–

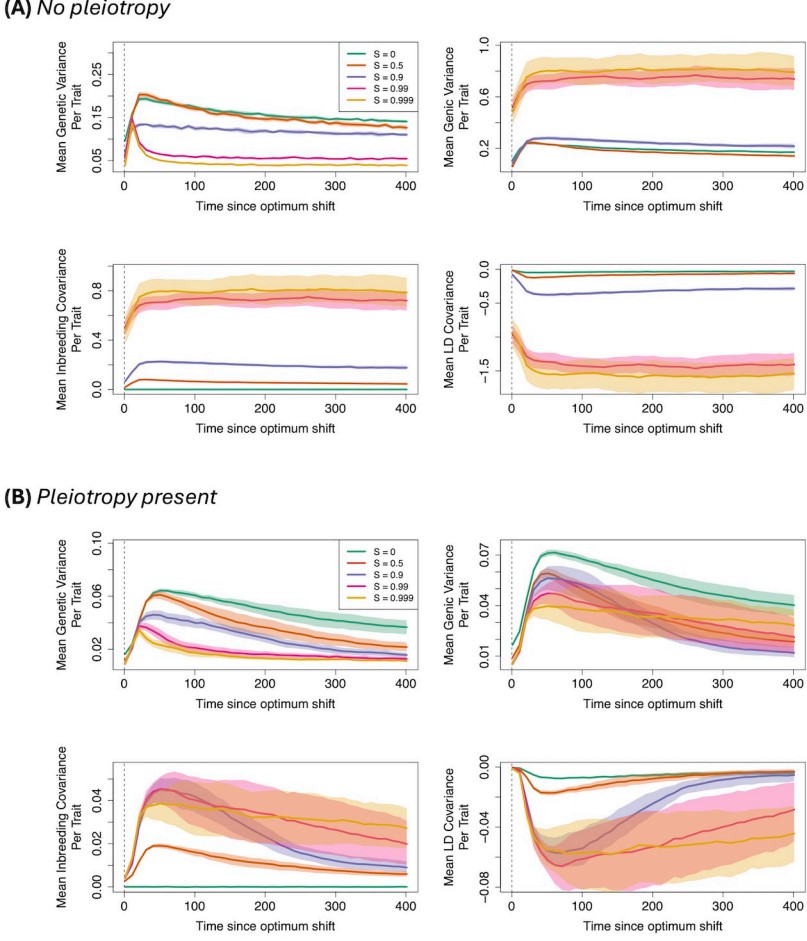

**Fig 2. Genetic variance under adaptation.** Genetic variance (top–left); genic variance (top–right); inbreeding covariance (bottom–left); and LD covariance (bottom–right) after an optimum shift. Different colours represent different fractions of self–fertilisation, as denoted by the plot legend. Solid lines are mean values, while bands denote 95% confidence intervals. Each trait mutation affected either 1 (A) or 5 (B) traits.

fertilisation values. These results reflect a process where large–effect mutations, both favourable and maladaptive, experience sudden frequency changes before reaching a new steady–state but not necessarily fix. However, with very high self–fertilisation (99% and above), variance instead spikes and subsequently reduces to zero, implying that following a rapid change in allele frequencies, the genetic variance available to selection quickly exhausts itself.

Under pleiotropy (Fig 2B), genetic variance quickly reaches a maximum then noticeably decreases after the optimum shift for all selfing values, showing how pleiotropy reduces the probability of polygenic selection occurring, and instead trait mutations are more likely to be fixed or lost rather than maintained at intermediate frequencies (as observed in previous models of polygenic selection, e.g., [64]). Note also that inbreeding and LD covariance sometimes take a long time to reach equilibrium values, even after the new optimum has been reached.

Our results for the different variance components are in qualitative agreement with our analytical results (Eq 12) and generally reflect those obtained by [52], who studied genetic variance in selfing species under adaptation. Both the genic and inbreeding variance increases with the degree of self–fertilisation, while the LD covariance is strongly negative. These results

show how trait variants are more likely to be present as homozygotes at individual sites under high selfing (as reflected by increased inbreeding covariance), while also being in high linkage disequilibrium, reflecting selection interference (Eq 12). Yet one key difference is that [52] found adaptation proceeds through a reassortment of cryptic variation under high self–fertilisation, caused by residual outcrossing creating new genetic combinations. This process is characterised by a spike in additive variance and a decrease in the magnitude of LD covariance just after an optimum shift. However, we do not observe this outcome in our results. There are several possible reasons for this discrepancy due to differences in modelling assumptions. These include that [52] assumed free recombination between sites, whereas linkage in our model could alter the selection response; stabilising selection is stricter in their model leading to negative LD covariance at equilibrium; cryptic reassortment occurred over very short timescales (20 generations); their optimum shift was much stronger ($z_1 = 2.5$, as opposed to no more than 1 here); or that [52] only investigated the effect of standing genetic variation.

**Haplotype structure.** *Trait variant frequencies.* Haplotype plots demonstrate how trait variants spread following the optimum shift (going from left to right in Fig 3). To determine the effect of the mating system, we compare results under the cases of complete outcrossing against high self–fertilisation (99.9%). With one trait and outcrossing (Fig 3A), we see the typical polygenic selection regime at work; there are many variants present in the population, and as adaptation proceeds then while some of them increase in frequency, we do not observe any particular mutation going to fixation via a selective sweep. Under high self–fertilisation (99.9%; Fig 3B) we observe a drastic change in genome structure. Before the optimum shift there are instead many selected variants at high frequency, with compensatory effects due to stabilising selection. As adaptation proceeds, we see other mutations segregating in the background and contributing to adaptation. These results reflect the analytical findings that alleles with opposite effects create positive LD (Eq 3) that subsequently enhances the spread of favourable alleles during evolution (Eq 6).

The presence of pleiotropy also affects these results. Under outcrossing (Fig 3C) then while the polygenic regime is still present, the density of mutations has been reduced. This reflects the known property that mutations are more likely to be deleterious with higher pleiotropy, so fewer are maintained in the population [51, 68]. Another major change occurs when both pleiotropy and self–fertilisation are present (Fig 3D); we observe a clear block–like structure within sampled genomes in line with a transition to genome–level selection (similar to [46]). As the optimum shifts, we see positive–effect variants go to fixation in line with a traditional selective sweep, but with apparent genome–wide hitchhiking of mostly neutral mutations with the occasional linked trait variant.

*Linkage disequilibrium.* We can investigate the transition to a more 'clonal'–like structure by plotting pairwise LD along the genome. Fig 4 shows examples of genome–wide heatmaps of LD for individual simulation replicates. In the outcrossing cases (Fig 4A and 4C) most LD values are very low, indicating that variants segregate independently over long genetic distances in line with traditional polygenic selection mechanisms. With high selfing and no pleiotropy (Fig 4B) we observe regions of high LD being maintained over parts of the genome, with distinct genetic 'blocks' forming as adaptation occurs. These results reflect both how selection affects larger linked regions of the genome, but also how LD is maintained under a sufficiently high selfing rate (Eq 7). When pleiotropy is added (Fig 4D) these LD blocks can extend across the genome as adaptation is occurring. Also note in this particular example is that LD blocks are formed long after the new optimum has been reached, in line with preceding results demonstrating that LD covariance can take long periods of time to equilibrate (Fig 2).

We can gain additional insight into how linkage blocks are formed by selfing by inspecting how LD changes as a function of distance. With outcrossing, LD is maintained at a near-zero

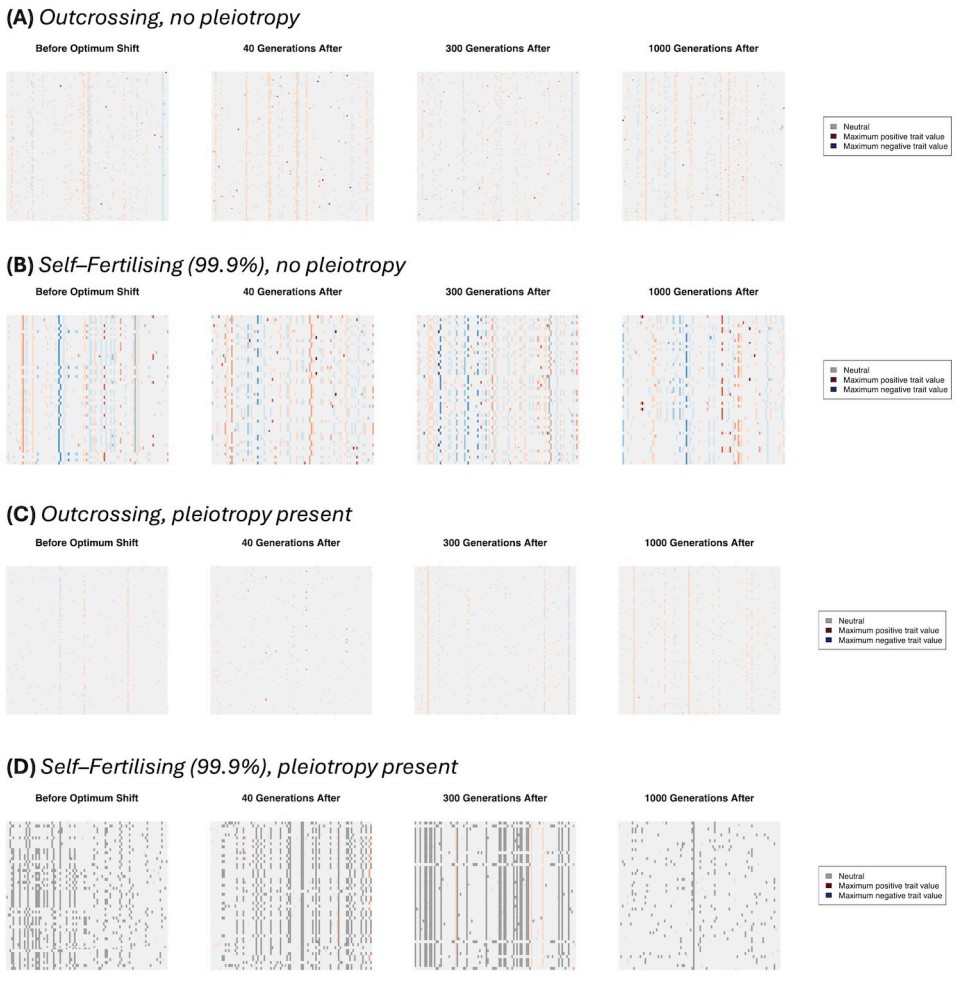

**Fig 3. Haplotype plots over several time-steps.** The X–axis correspond to genomic positions along the chromosome and the Y–axes to sampled individuals, taken at different times at denoted above each plot. Different colours represent mutation types; black points denote neutral variants, red points are trait mutations with a positive effect when averaged over all traits, and blue points are trait mutations with a negative–average effect. Darker colours denote a larger magnitude of average effect. Note that we do not show trait mutations that have fixed in the sample to aid visualisation. Populations are either outcrossing (A, C) or self–fertilising with $S = 0.999$ (B, D). Mutations either affect 1 trait (A, B) or 5 traits (C, D).

level (Fig 5A), indicating little LD between SNPs (for this sampling level of 0.5Mb apart). Conversely, under very high self–fertilisation without pleiotropy, LD is maintained at high levels (between 0.25–0.75) before the optimum shift and continue at these high levels as adaptation is ongoing (Fig 5B). When pleiotropy is present, then while LD is also initially at a high value, it noticeably elevates as adaptation proceeds before reducing back towards pre-adaptation levels (Fig 5C).

There is also a wide variation in outcomes over individual simulation replicates, potentially indicating that the process of adaptation is greatly different between simulation runs. However, LD measurements are known to be affected by the underlying allele frequencies, even in measurements like $r^2$ that correct for them [69, 70]. We hence also investigated LD decay using the $|D'|$ statistic, which while still being affected by the underlying allele frequencies, normalises LD by the maximum possible value so it can yield insight into to what extent the range of possible LD values are present. In this case, $|D'|$ values are close to one over the observed

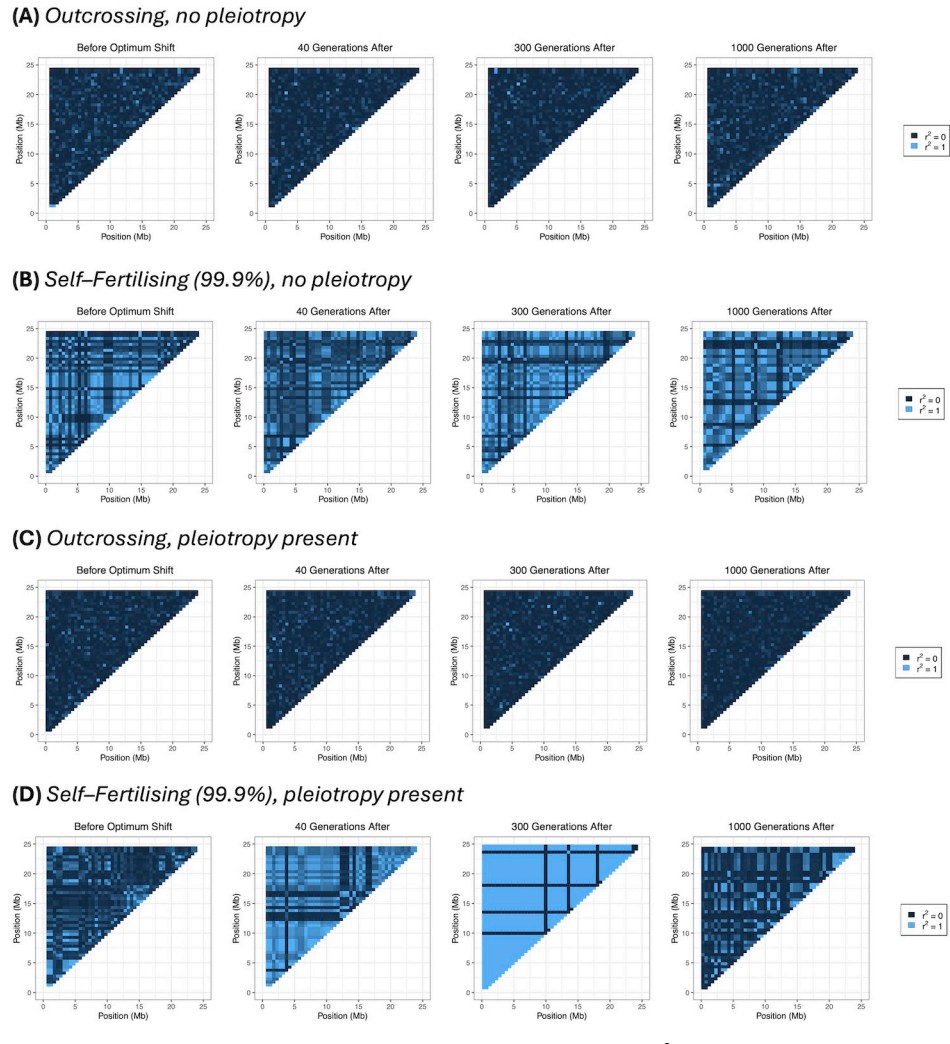

**Fig 4. Heatmaps showing linkage disequilibrium (LD).** LD is measured by $r^2$ across the sampled genomes and time points shown in Fig 3. Colours range from dark blue representing $r^2 = 0$, to light blue representing $r^2 = 1$.

distances in the 99.9% self–fertilisation case (Fig A in S3 File). Hence, most variants over a genome are in extremely tight linkage, with variations in $r^2$ reflecting differences in the underlying allele frequencies.

*Change in allele frequencies.* We next investigate the change in allele frequencies over each timepoint, to determine the nature of adaptation under each scenario and how it is affected by the mating system. Note that we only track mutations that are present just before the optimum shift (see Methods for further details). With outcrossing and no pleiotropy (Fig 6A), we recover the polygenic regime with many selected alleles changing in frequency, with stronger–effect variants are usually maintained at very low frequencies. In contrast, under high selfing (Fig 6B), we see co–ordinated allele frequency changes with sets of mutations changing frequencies together, with sets of mutations having opposing mutation effects. The allele–frequency changes are also noticeably sharper than under outcrossing, but not in a single direction. These results reflect analytical findings, demonstrating how selfing causes alleles with contrasting effects to change frequency due to LD, but negative LD does not lead to straightforward directional selection (Eqs 6 and 7).

**(A)** *Outcrossing, no pleiotropy*

**(B)** *Self–Fertilising (99.9%), no pleiotropy*

**(C)** *Self–Fertilising (99.9%), pleiotropy present*

**Fig 5. LD decay over the simulated genome.** LD, as measured by $r^2$, as a function of distance (note that only distances up to 12.5Mb, covering half the simulated genome, are considered). Different colours represent different simulation replicates, with points being individual samples, and lines representing the mean value over distance (bands are 95% confidence intervals). The black dashed line shows the mean value over all simulations. The following cases are plotted: (A) outcrossing, no pleiotropy; (B) 99.9% selfing, no pleiotropy; (C) 99.9% selfing, pleiotropy present.

When pleiotropy is added, the transition to individual–mutation effects driving adaptation becomes more noticeable. Under outcrossing (Fig 6C), we see that only a couple of mutations present at the optimum shift sweep to fixation, while the remaining mutations go extinct. In contrast, no mutations that are present at the optimum shift fix under self–fertilsation when pleiotropy is present (Fig 6D). These results further demonstrate how pleiotropy leads to a transition from polygenic selection to monogenic selection.

**Gradual optimum shift.** We also investigated simulations where there is a gradual change in the optimum, as opposed to a sudden change. Results are outlined in Figs B–D in S3 File; these are generally the same as for the instant–shift case, with the main exception being that the increase in mean trait value due to selection occurs later after the optimum starts changing.

## Larger population size

To check if these results are robust to changes in population size, we re-ran the neutral, non–pleiotropic simulations with an elevated population size of $N = 10,000$. Results are outlined in Figs E–G in S3 File, and we see that the same general patterns are obtained.

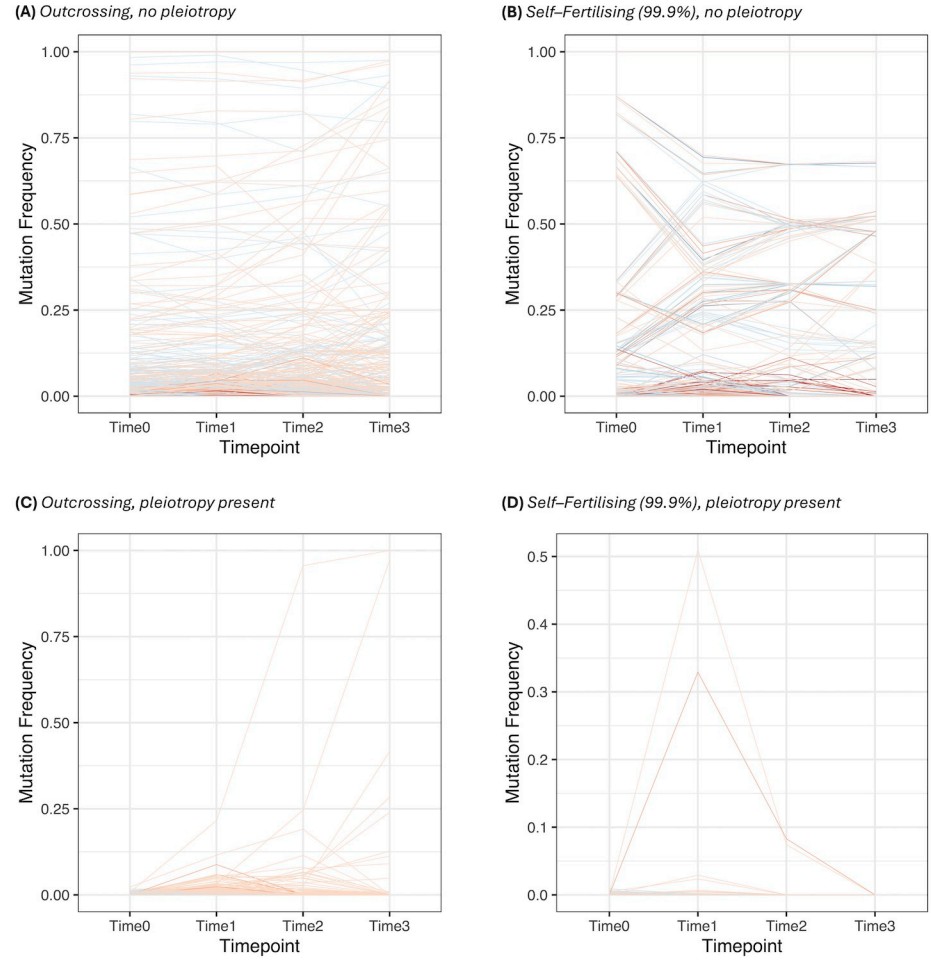

**Fig 6. Plots of derived allele frequencies over the four timepoints that were previously used to plot haplotype diversity.** The colours denote whether the trait value is positive or negative, with darker colours denoting stronger–effect mutations (as in Fig 3). The following cases are plotted: (A) outcrossing, no pleiotropy; (B) outcrossing, pleiotropy present; (C) 99.9% selfing, no pleiotropy; (D) 99.9% selfing, pleiotropy present.

## Including background selection

Deleterious mutations are prevalent in nature [71, 72], with empirical and theoretical work estimating that they have recessive effects on average [73]. Recessive deleterious mutations are hypothesised to be a major contributor to inbreeding depression [74, 75], and could affect the role of polygenic adaptation through influencing trait variants via selective interference. We hence added recessive deleterious mutations to the simulations with dominance coefficient $h = 0.2$, to determine how they affect polygenic adaptation and the formation of linkage blocks.

In this case, the fitness of highly–selfing populations is reduced and becomes closer to that for 90% selfing, despite exhibiting the lowest variance in fitness (Fig 7A). This demonstrates how the presence of recessive deleterious mutations that are exposed to selection under selfing reduces the benefits of the mating system in strengthening stabilising selection. Inbreeding depression reduces with increased selfing, demonstrating how these deleterious mutations are purged from the population. The quantitative genetic variance measurements do not greatly

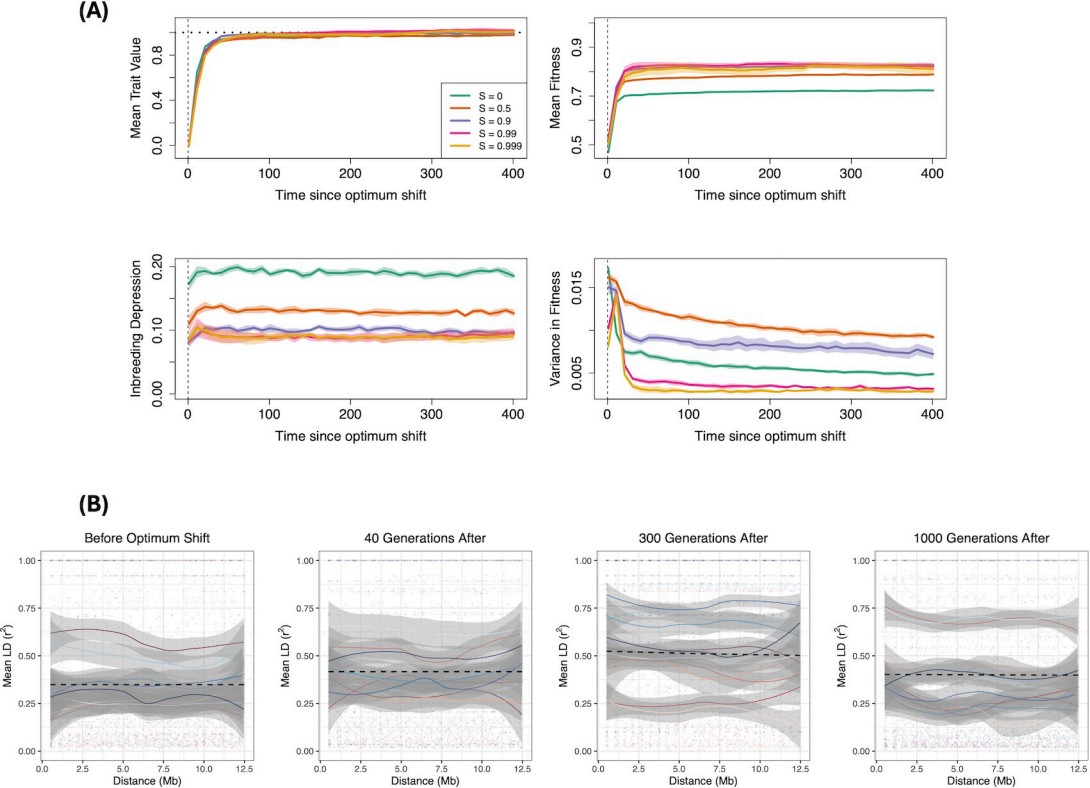

**Fig 7. Results when background deleterious mutations are included.** (A) Mean trait value, fitness, inbreeding depression and variance in fitness for different rates of selfing when background deleterious mutations are also included, with dominance coefficient $h = 0.2$. (B) LD decay of population samples under 99.9% selfing. Trait mutations are non–pleiotropic.

differ compared to the case without deleterious mutations (compare Figs 2A and H in S3 File). One key difference is that the variance components for the highest selfing case revert back to zero over time, possibly indicating that the presence of deleterious mutations breaks down the formation of linkage blocks, and instead leads to adaptation via individual sweeps. This is indeed what we observe from trait allele–frequency changes over the optimum shift (Fig I in S3 File).

The idea that deleterious mutations prevents the formation of linkage blocks under high selfing is reinforced by looking at LD decay (Fig 7B). Under 99.9% selfing there are a lower variance in values over time; in particular, there are fewer high–LD cases at later timepoints (300 generations and afterwards), unlike results without deleterious mutations. Overall, these results indicate that background selection disrupt adaptation in highly–selfing species, so fewer trait mutations fix together. Fig J in S3 File provide examples of haplotype plots and LD heatmaps for individual simulation runs.

## Comparing the effects of self–fertilisation and low recombination in outcrossing species

The formation of linkage blocks under self–fertilisation is a consequence of increased selfing reducing the efficacy of recombination due to the long runs of homozygosity formed under inbreeding (see also Eq 7). Similar linkage effects could also arise in outcrossing genomes with a low overall recombination rate. To determine to what extent observed results reflect altered

effective recombination and mutation rates under self–fertilisation, we compared the highly selfing results to an outcrossing population with rescaled recombination and mutation rates to reflect their effective values under high selfing ([34, 57]; see Methods for further details).

Results are outlined in S3 File. We demonstrate that the genetic variance components are different between the two cases; in particular, there is an elevated inbreeding and stronger negative covariance in the highly–selfing case. When recessive deleterious mutations are added, then low–recombination outcrossing populations exhibit stronger negative LD covariance, reflecting how selective interference is more prevalent in this scenario [76, 77]. Finally, LD generally decays over shorter distances in outcrossing populations than in highly–selfing cases, although some long–range LD can be formed if deleterious mutations are present. Hence long–range LD in selfing species is also a consequence of increased homozygosity.

## Discussion

### Summary of results

Self–fertilisation, like any form of inbreeding, changes how genetic variants are inherited between parents and offspring, which also affects how diversity is structured. These changes, including increased homozygosity and reduced effective recombination, also affects how polygenic selection is realised. Here, we combined analytical modelling with a comprehensive set of simulations to investigate how polygenic adaptation is affected by self–fertilisation. In particular, we demonstrate how high self–fertilisation changes the polygenic selection response.

We first outlined two-locus results to show how the selection response, and formation of linkage disequilibrium, is affected under self–fertilisation. We first showed that selection at the optimum will cause LD to be negative if selected alleles are aligned ($\gamma_A\gamma_B > 0$) and positive if opposed ($\gamma_A\gamma_B < 0$) (Eqs 3 and 5). As an optimum shifts, we further showed that if selfing is sufficiently high and/or recombination sufficiently low then the magnitude of LD will be maintained over time (Eq 7). We also showed that selfing has two opposing effects on changing the frequency of selected alleles; it increases their rate of spread due to stronger direct selection, but can also incur negative LD that can impede their spread (Eq 6).

Next, using simulations to determine which of these phenomena most strongly affects the selection response, we observed that self–fertilisation has a greatest influence on the long–term effects of genetic adaptation once an optimum has been reached. The mating–system has little influence on the time needed for the population to reach its new optimum, both in the simplest case where only trait variants are present (Fig 1) and also when deleterious alleles are present (Fig 7A). This result demonstrates how the contrasting effects of selfing demonstrated in Eq 6 cancel out leading to a similar rate of adaptation, at least under the conditions we explored. However, once the optimum has been reached then highly–selfing populations exhibit the highest fitness and lowest inbreeding depression (Fig 1) due to more direct selection acting on trait mutations (Eq 6). Similar results were obtained by [53] in the context of tracking a population invading a new region; highly–selfing populations were more able to establish if the source population is also highly selfing and was able to purge its inbreeding depression.

By inspecting genetic variance and haplotype structure (Figs 2–5), we observe that once selfing becomes sufficiently high, it creates genome–wide linked blocks of trait variants consisting of compensatory mutations (Eq 7). If mutations are pleiotropic then we observe a transition from a polygenic–selection regime to a monogenic selection regime, where only a few large–effect variants go to fixation. LD is extensive in this case, but it seems that many hitch-hiking mutations are neutral. We also see these effects influence allele frequencies, with co–ordinated change in linked alleles under high selfing, but only a few trait mutations sweep when pleiotropy is added (Fig 6). Note that the transition to coordinated allele changes

occurred at very high selfing rate in our simulations. However, we used a rather low genomic mutation rate on trait mutations (less than 0.01). It is likely that the transition would occur at lower selfing rate with higher mutation rate (see also [51]), but this is currently hard to investigate due to the increased memory load this will incur in these simulations.

The presence of recessive deleterious mutations removes the long–term fitness advantage that highly self–fertilising organisms otherwise obtain under stabilising selection, likely because many deleterious mutation are exposed to selection (Fig 7A). Their presence also weakens the LD structure that is otherwise formed during polygenic adaptation, with fewer high–LD populations present after the optimum has been reached (Fig 7). These results are similar to those observed by [54] in a related model of local adaptation under self–fertilisation, where the addition of deleterious mutations lead to clustering of variants that underlie local adaptation over shorter physical distances than in simulations when they were absent.

Finally, we compared the high–selfing results to an outcrossing model with rescaled recombination and mutation rates, as both will cause the maintenance of LD as adaptation is ongoing. However, the two cases do not yield equivalent results (details in S3 File). Generally, while both cases show the effects of selection interference, selfing populations also exhibits elevated inbreeding covariance reflecting heightened homozygosity driving evolution of trait variation. We also see a prevalence of selection interference in the outcrossing case when deleterious mutations are added. Long–range LD is still present under the rescaled outcrossing case but is not as strong as under the selfing case, but can be formed when deleterious mutations are present. Overall, these results suggest that it is not just the reduced effects of recombination causing a transition to genome–level selection, but increased homozygosity also plays an important role.

## Implications for genetic evolution and genome inference

One of the earliest predictions regarding adaptation in self–fertilising species is that selfers should be less able to adapt to their environment, compared to outcrossing species [38]. Our findings run contrary to these early, verbal predictions. The mating system has little effect on how long it takes species to adapt to new environments; once the new optimum is reached, then the highest fitness is usually observed in species with a high degree of self–fertilisation. These advantages seem to arise due to the purging effect of selfing, which removes both large–effect variants that lead to maladaptation, along with recessive deleterious mutations. Other theoretical studies have also shown advantages to self–fertilisation when adapting to new environments [52, 53], highlighting how it does not always impedes adaptive potential. However, negative effects of selfing can adversely affect long–term population fitness (see 'Open Questions' below).

Traditional models of polygenic adaptation, and methods for detecting it, generally assume that individuals are idealised outcrossers and that polygenic variants segregate independently from one another. The results presented in this manuscript suggest that these assumptions are violated under high levels of self–fertilisation (typically above 90% in these simulations), and low recombination rates. Clusters of selected mutations in high linkage disequilibrium are likely to form; in extreme cases large regions of the genome forming a block consisting of many trait variants in linkage with one another. Furthermore, the presence of pleiotropy causes many selected variants to be purged, meaning that adaptation is more likely to proceed via the fixation of fewer major–effect mutations (Figs 2 and 3).

Given that blocks are more likely to form during polygenic selection under self–fertilisation, then their presence needs to be accounted for when testing for it in genome studies. Moreover, this block structure can persist long after the macroscopic characteristics (e.g.,

mean fitness, genetic variance) have reached the new equilibrium (compare Figs 1 and 4). Empirical polygenic selection studies, especially in humans, usually consider a single variant per linked region to correct for linkage effects [78–81]. The findings in the study urge caution with this approach, as linkage can cause other selected alleles to fix in their vicinity and hence might affect the realised trait effect of these mutations. Future work should further investigate what impact this approach may have on polygenic selection studies. Our findings also show that it will be even more important to account for regions of high linkage before performing polygenic selection tests in self–fertilising organisms. LD, as measured by $r^2$, is a standard population–genetic statistic for identifying linked regions and so can be readily used in the first instance to identify co–linked regions. In addition, these results show that these outcomes are not just a consequence of reduced recombination in outcrossers. Self–fertilisation also leads to purging of deleterious mutations and extreme trait genotypes, as exemplified by reduced inbreeding depression (Figs 1 and 7A). Future research can aim to disentangle the roles of reduced recombination and increased homozygosity in forming linked clusters.

Another challenge with analysing data from self–fertilising species, as highlighted by this study, is subsequently identifying what selective mutations are present in each block; whether there are many selected variants or just one variant with many hitchhikers. A first test could be to measure the length of swept haplotype blocks; longer regions are more likely to contain functional variants that contribute to polygenic selection, but such a test does not fully resolve the nature of selection within these haplotypes (e.g., whether mutations are major- or minor-effects, or with compensatory effects). Given that pleiotropy is associated with a lower density of adaptive trait variants and more neutral hitchhikers, then measuring the pleiotropy of each gene in a block (through, for example, measuring how connected genes are in a regulatory network; [82–84]) could provide information on whether adaptation is driven by polygenic or monogenic adaptation. Analysis of further gene information (e.g., looking at prevalence of non–synonymous variants or genes with known functions) can then shed light on possible selection mechanisms.

## Open questions

A major question arising from this work is to what extent neutral diversity actually reflects truly neutral processes, given that linked selection can be extensive during polygenic adaptation. For example, linkage disequilibrium covariance takes a long time to equilibrate after the optimum is reached (Fig 2) and can even be maintained if selfing is sufficiently high (Eq 7), which appears to result in long-distance LD formed many generations after the population reaches the new optimum (Fig 4). In evolutionary genetic studies, it is often important to separate out neutral genome regions from selected sites, but if LD is extensive then analyses of these putative neutral sites can be affected by selection. One example is estimating a population's demographic history from genome data, which can be misinferred if neutral regions have been affected by linked selection [85]. Highly–selfing species can hence be affected by these linkage effects [e.g., *Caenorhabditis elegans*, which exhibits a self–fertilisation rate of at least 99% [86] and for which chromosome–scale selective sweeps have been detected [45]]. Similarly, LD–based approaches to infer population–scale recombination rates should be interpreted with caution in high selfers, as inferred recombination landscapes could also reflect selection landscapes. In these cases it will be important to filter genome data over longer regions to ensure putatively neutral SNPs are not in linkage disequilibrium, otherwise it may be necessary to jointly account for selection and demography [87]. If time–series data were available, one could measure synonymous diversity before and after adaptation to directly measure to what extent neutral diversity is affected by linked selection.

These findings also shed new light into how the mating–system, mutation rates and pleiotropy affect genetic variance under inbreeding (reviewed by [88]). The earliest models concluded that inbreeding would increase additive variance in the absence of selection [89, 90]. [91] subsequently showed that while stabilising selection would reduce genetic variance, the mating–system has little effect on the variance maintained in a population. Subsequent work has clarified how self–fertilisation does decrease genetic variance, but there is debate over whether this reduction is gradual [51, 92] or drastically reduced ('purged') above a threshold selfing rate [50]. Our results show a mixture of both outcomes, depending on the pleiotropy present. In the simplest model (no background deleterious mutation or pleiotropy; Fig 2A) genetic variance is greatly reduced when selfing becomes really high, in line with the purging result of [50]. However, when pleiotropy is present (Fig 2B) then genetic variance reduces more gradually with the selfing rate, in line with the models of [92] and [51]. This transition was likely caused by pleiotropy reducing the trait mutations present, so there was less trait variation to purge. However, additive variance was also elevated with high selfing when pleiotropy was absent, with strongly negative LD covariance reducing the overall genetic variance (see also [52]). These results hint that the genetic variance present in selfers is affected by many factors, and elucidating how these all affect the long–term response to selection will be a rich area of future research.

Throughout, we have focussed on the creation of linked genotypes as measured by linkage disequilibrium (principally $r^2$ but also using $|D'|$). In mixed-mating populations, different levels of self–fertilisation in a population lead to 'identity disequilibrium', characterised by long runs of homozygosity [47, 93]. Building on models by [48, 49, 50] used selfing age–classes in quantitative genetics models to account for identity–disequilibrium, so it is a likely cause of the variance purging they observe. We hence foresee that investigating the joint role of linkage and identity disequilibria will prove useful when extending this model to consider variation in self–fertilisation in a population.

The results could also change with the modelling assumptions. We assumed 'universal pleiotropy' where each variant affected all traits simultaneously [95]. Partial pleiotropy, where only a few traits are affected by each mutation, could change the outcome of these simulations. However, since some pleiotropy will be present then we expect similar results to arise (i.e., monogenic selection with long–range linkage effects under high self–fertilisation). Fitness effects of mutations can also change if we alter the fitness landscape that determines the optimum [68]. We also assumed a genome structure consisting of alternating genic and non–genic regions; the size and distribution of these could also affect results through changing the prevalence of trait variants and deleterious mutations. Finally, we only consider additive effects on the trait mutations; the presence of non–additive effects can change the dynamics of adaptation due to a change in difference variance components following a shift to self–fertilisation [96].

A major open question is with regards to the long–term fate of outcrossing and self–fertilising species. In these simulations, populations with highest fitness after reaching the new optimum tended to be those with very high levels of self–fertilisation, and these advantages are not completely removed in the presence of deleterious mutations. Similar results were observed by [53] who showed that the purging of inbreeding depression in selfers may aid colonization and adaptation to a new habitat. Thus, available standing genetic variation does not appear as a limiting factor for adaptation in selfers (see also [52]), which is also in agreement with a meta–analysis showing limited effect of mating system on heritability [97]. However, once the new optimum is reached then there is a risk that highly selfing populations could go extinct due to weakened selection against deleterious mutations [98–100]. Furthermore, selfing not only alters the genetic functioning of a species but is also expected to strongly affect its ecology and

population dynamics, through creating more frequent bottlenecks along with smaller and more ephemeral populations [101]. It is feasible that these demographic effects interact with linked selection, lowering the long–term adaptive potential of selfing species. It remains to be tested whether these theoretical short–term advantages are present in nature and whether they are sufficient to maintain self–fertilising populations, and whether they interact with demography to create a long–term disadvantage to selfing lineages.

## Methods

### Simulation procedure

Simulations were ran using SLiM version 3.3.2 [66]. For most simulations, a diploid population of 5, 000 individuals was simulated. We used a per–nucleotide mutation rate of $\mu = 4 \times 10^{-8}$; although this is approximately six–fold higher than that measured in *Arabidopsis thaliana* ($7 \times 10^{-9}$; [102]), the estimated $N_e$ for *A. thaliana* populations from southern Sweden is 198,400, which is nearly 40-fold higher than that used here, so the net input of new mutations (as measured by $N_e\mu$) will actually be lower than expected in wild populations. This mutation rate was kept throughout the simulation (i.e., both during the burn-in phase and after the change in mean optimum). The recombination rate was set to $\sim 1.98 \times 10^{-7}$; this value was obtained by converting the estimated genomic recombination rate of 3.6cM/Mb [103] to a per-nucleotide rate using Haldane's mapping formula [104], then rescaling so that the mutation–to–recombination rate ratio was the same as empirical estimates for *A. thaliana*. The 'mutation stacking' policy was set so that only the first–appearing mutation at a site was retained.

Each individual had a genetic architecture similar to that used by [54]. There was a single chromosome of length 25Mb long, consisting of 4000bp–long neutral buffer space and 1000bp–long gene regions. All mutations in buffer regions were neutral. In genes, 25% of mutations were neutral; 65% of mutations are deleterious, and 10% affect the quantitative trait. The neutral fraction reflects the number of changes affecting presumed–neutral synonymous sites [54]. With this setup, an average of 100 new trait mutations will be introduced into the population every generation. In a theoretical analysis of polygenic adaptation, [12] showed that this level of mutation input will lead to polygenic adaptation via subtle frequency shifts; it has subsequently been shown that this condition holds generally across different fitness models [14, 105]. The frequency of self–fertilisation was set to one of 0, 0.5, 0.9, 0.99, or 0.999; these values were set using the 'setSelfingRate' function in SLiM. The three first values allow to compare outcrossing, mixed mating and high selfing, whereas the two last ones focus on the presumed critical zone between high and very high selfing rates where clonal genotypes are likely to form.

We investigated cases where the only selected mutations were those that affected the trait, or where a fraction of mutations were uniformly deleterious. If present, these deleterious mutations had a selection coefficient $s = -0.02$, and a dominance coefficient of $h = 0.2$. Otherwise, we set $s = 0$ to disable their presence. Values were set using the 'initializeMutationType' function in SLiM. Each trait mutation affected $n$ sub–traits, which we either set to 1 (no pleiotropy) or 5 (pleiotropy present). For $n = 1$, mutation effect sizes were drawn from a normal distribution with mean 0 and standard deviation $\sigma = 0.25$. If $n > 1$ then effect sizes were drawn from a multinomial distribution, where each trait had mean 0 and variance $(0.25)^2/n$. When pleiotropy is present, the mean selection coefficient of mutations is proportional to $n\sigma^2$ [68]; rescaling the trait variance by $n$ will therefore enable the mean selective effects to be approximately the same as in the univariate case, enabling us to test for the effect of pleiotropy alone. Trait effects were additive within– and between loci;

hence, homozygote mutations had twice the effect size as heterozygotes, and the value of trait $i$ within an individual, $z_i$, is summed over all mutations.

Individuals' fitness were calculated from the product of those contributed by both background deleterious mutations and trait mutations. The fitness contributed by deleterious mutations is $w_d = (1 - hs)^{nhet}(1 - s)^{nhom}$ for $nhet$ the number of heterozygote sites and $nhom$ the number of homozygote sites. The fitness contributed by trait mutations, $w_q$, is calculated over all $n$ traits using a multivariate Gaussian fitness function:

$$w_q = \exp\left( -\frac{1}{2} \sum_{i=1}^{n} \frac{(z_i - z_0)^2}{V_s} \right) \tag{13}$$

where $z_0$ is the optimal trait value, and $V_s$ is the variance in the Gaussian fitness curve, which determines how strict fitness is. $V_s$ is set to 1 throughout. This part of an individual's fitness was calculated using a custom fitness callback in the SLiM script. Note that although mutation effects are additive within and between loci, stabilising selection naturally induces dominance and epistasis on fitness, although mean epistasis at the optimum is zero under this fitness function [68, 73, 106].

Simulations were ran for a a burn–in period of $10N = 50,000$ generations with $z_0 = 0$ for all traits. In the main simulation set, the optimum was subsequently and instantaneously changed to $z_0 = 1/\sqrt{n}$; this new value ensures that the initial drop in fitness due to maladaptation is the same irrespective of $n$ [68]. Plotting the mean fitness and genetic variance over the burn–in period shows that the population has reached a steady–state by the end of this period (see Fig Q in S3 File for an example).

We also ran simulations where the optimum gradually changed from 0 to $1/\sqrt{n}$ over the first 100 generations after the burn–in, with the optimum increasing by $1/(100\sqrt{n})$ each time. All simulations were ran for 1001 generations following the burn–in to enable the population to adapt to the new optimum.

To test the generality of our results to the population size, we re-ran the set of simulations without deleterious mutations and pleiotropy with a larger population size of $N = 10,000$. In these cases the burn–in period was kept at 50,000 generations, i.e., $5N$ generations, due to memory limitations. All other parameters were unchanged.

We also simulated an outcrossing population but with rescaled parameters to match those expected under self–fertilisation, to test whether the observed results are caused principally by the reduced recombination incurred. In these simulations, the mutation rate was divided by $(1 + F)$ for $F$ the inbreeding coefficient, which equals $\sigma/(2 - \sigma) \approx 0.998$ for a selfing fraction $\sigma = 0.999$. This rescaling reflects the reduction in effective population size $N_e$ caused by selfing, and hence the effect on the net introduction of mutations $N_e \mu$ [34]. Rescaling the recombination rate is tricker; for a genome tract with total recombination rate $r$ acting over it, the effective recombination rate under selfing equals $r(1 - 2F + \Phi)$, where $\Phi$ is the probability of joint identity–by–descent at two loci and is a function of both the selfing and recombination rates; $\Phi \approx F$ unless self–fertilisation and recombination rates are both large [57]. By default, SLiM defines recombination rates on a per–basepair basis, but defining $\Phi$ using this value is ineffective if we want to rescale recombination across the whole genome. To this end, we rescaled the recombination rate using $\Phi$ defined using $r = 1/2$; the justification here is that the mean number of breakpoints per individual, $r(L - 1)$ for $L$ the number of basepairs (25Mb), will subsequently be rescaled by $(1 - 2F + \Phi)$. In this case, $\Phi = [\sigma(2 + \sigma)]/[(4 - \sigma)(2 - \sigma)] \approx 0.997$ for $\sigma = 0.999$, and the rescaled recombination rate approximately equals $2.65 \times 10^{-10}$.

## Calculating statistics

All statistics, except haplotype diversity and linkage disequilibrium measurements, were outputted every 500 generations before the optimum shift, or 10 generations after the optimum shift. We were interested in capturing three broad patterns of adaptation: (i) the change in fitness over time; (ii) the genetic variance over time; (iii) haplotype structure, and how many trait variants contribute to adapted genotypes.

To measure fitness effects, we outputted the population mean trait value; the mean and variance in fitness; and inbreeding depression. The latter was calculated by creating two new sub-populations, each of size 500, where one was obligately outcrossing and one was obligately self–fertilising. Inbreeding depression was subsequently calculated as $1 - (w_S/w_O)$, for $w_S$ the mean fitness of the self–fertilising subpopulation and $w_O$ the mean fitness of the outcrossing populations. Mutation rates were also set to zero in each subpopulation, so fitness calculations were based on variation that was only present in the main population. Both subpopulations were subsequently removed until the next check.

We also calculated the genetic variance, and decomposed it to determine which factors most strongly contribute to adaptation (a similar approach was used by [52]). Let $p_j$ denote the frequency of trait mutation $j$; $f_{0,j}, f_{1,j}$ and $f_{2,j}$ be the frequency of wild-type homozygotes, heterozygotes, and derived homozygotes; and $a_{i,j}$ the effect size of mutation $j$ affecting trait $i$. Then we calculate the genetic variance of trait $i$, $V_{G,i}$, as:

$$V_{G,i} = \sum_j (f_{1,j}a_{i,j}^2 + f_{2,j}(2a_{i,j})^2) - \mu_i \tag{14}$$

$$\text{for mean } \mu_i = \sum_j (f_{1,j}a_{i,j} + f_{2,j}(2a_{i,j})) \tag{15}$$

We further decompose the genetic variance into the genic variance $V_g$, which is the variance assuming Hardy–Weinberg expectations and no linkage; the inbreeding covariance $V_I$ reflecting deviations from Hardy–Weinberg equilibria; and the linkage covariance $C_{LD}$. These terms were calculated using equations provided in [65], modified to take different effect sizes into account:

$$V_{g,i} = \sum_j 2p_j(1 - p_j)a_{i,j}^2 \tag{16}$$

$$V_{I,i} = \sum_j (2f_{2,j}f_{0,j} - (1/2)f_{1,j})a_{i,j}^2 \tag{17}$$

$$C_{LD,i} = V_{G,i} - V_{g,i} - C_{I,i} \tag{18}$$

We plot the mean of the variance values over all traits. To check the simulation code, we tested whether the pre–shift genic variance $V_g$ matched the house–of–cards expectation, $4U_tV_s$ for $U_t$ the total mutation–rate of trait variants [107], in the simplest case of outcrossing, no pleiotropy or background deleterious mutations. Given the genome setup and mutation rates, $V_g = 0.08$ for default parameters. We observed that the actual $V_g$ exceeded this expectation (Fig R in S3 File), which is a known mis–match if mutation rates are high [51]. However, if we reduced the mutation and recombination–rates 10–fold so expected $V_g = 0.008$, then the simulations did match this value. We hence conclude that the simulation seems to work as expected. 95% confidence intervals for fitness and variance measurements were calculated using 1,000 bootstrap replicates.

Haplotype plots were produced for the first simulation replicate of each parameter set. 50 individuals were sampled either the generation before the optimum shift if there was a burn–in; 40 generations after the optimum shift; 300 generations afterwards; and 1,000 generations afterwards. Haplotype information was printed out as a VCF file, along with information on the effect size of each trait variant. The number of mutations in haplotype plots were thinned to 100; if there were less than 100 trait mutations, then haplotype plots consist of a subset of segregating trait variants, then a sample of background mutations, both deleterious and neutral, if space allowed. We do not show trait mutations that have fixed in the sample of individuals.

Linkage disequilibrium (hereafter LD), as measured using both $r^2$ [108] and $|D'|$ [109], was calculated using *VCFtools* [110] after thinning SNPs so that variants are at least 0.5Mb apart and only including those with a minor allele frequency of 0.1. Results were plotted in two ways: first as a heatmap to demonstrate genome-wide clustering of mutants for a single simulation replicate, and as a function of distance between two alleles to show the decay of LD over the genome for all simulation replicates. In the latter case, we follow a similar procedure as in [111]; we first grouped measurements in bins of 0.5Mb long, based on the distance between the two SNPs. We then only consider distances less than 12.5Mb as measurements become too noisy otherwise due to fewer comparisons. We then thinned the number of entries to equal the smallest number over these remaining bins, to further reduce the effect of within-bin stochasticity. If the smallest number of entries in these bins is less than 10, we omitted that simulation replicate from the plot.

Allele frequencies were obtained from each of the four timepoints in the simulation; in this case, we obtained all trait alleles, even those not present in the sample of 50 individuals used to plot haplotypes and LD heatmaps. Alleles were only retained if they were present in the population just before the optimum shift.

## Other software used

Data was processed and plotted in R 3.6.1 [112]. We also used the following R packages: *cowplot* [113]; *gplots* [114], *plyr* [115], *RColorBrewer* [116], *ggplot* [117] and *tidyverse* [118]. Output PDFs were combined into single files using *pdfjam* (https://github.com/DavidFirth/pdfjam).

## Supporting information

**S1 File. *Mathematica* derivations, two–locus model.** *Mathematica* file of analytical derivations for the two–locus analysis.
(NB)

**S2 File. *Mathematica* derivations, multi–locus model.** *Mathematica* file of analytical derivations for the multi–locus analysis.
(NB)

**S3 File. Additional results.** File of additional simulation results.
(PDF)

**S4 File. *Mathematica* derivations, two–locus model; PDF printout.** PDF printout of S1 File.
(PDF)

**S5 File. *Mathematica* derivations, multi–locus model; PDF printout.** PDF printout of S2 File.
(PDF)

## Acknowledgments

We would like to thank Josselin Clo for helping interpreting results.

## Author Contributions

**Conceptualization:** Matthew Hartfield.

**Data curation:** Matthew Hartfield.

**Formal analysis:** Matthew Hartfield, Sylvain Glémin.

**Funding acquisition:** Matthew Hartfield.

**Investigation:** Matthew Hartfield, Sylvain Glémin.

**Methodology:** Matthew Hartfield, Sylvain Glémin.

**Project administration:** Matthew Hartfield.

**Software:** Matthew Hartfield.

**Validation:** Matthew Hartfield, Sylvain Glémin.

**Writing – original draft:** Matthew Hartfield.

**Writing – review & editing:** Matthew Hartfield, Sylvain Glémin.

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
