## [Decision Letter · Decision Letter 0]

21 May 2024

Dear Dr Hartfield,

We are pleased to inform you that your manuscript entitled "Polygenic selection to a changing optimum under self--fertilisation" has been editorially accepted for publication in PLOS Genetics. Congratulations!

You will note that one of the reviewers (#2) had some more substantive comments for you to consider, and you should do so before you submit the final version. But we are not conditioning acceptance on any changes.

Yours sincerely,

Rodney Mauricio, Ph.D.

Academic Editor

PLOS Genetics

Kelly Dyer

Section Editor

PLOS Genetics

Comments from the reviewers (if applicable):

Reviewer's Responses to Questions

**Comments to the Authors:**

Reviewer #1: The authors have given careful consideration to my comments on the first draft. I am satisfied the paper is ready to be accepted.

Reviewer #2: In this manuscript, Hartfield & Glémin investigated how adaptation occurs in populations with different levels of selfing. They did not only follow how fitness and variance change over time as in previous conceptually similar models (as in Clo et al. 2020 or Sachdeva 2019 for example), but they also deeply investigated how adaptation occurs and how the signature of selection is modified by self-fertilization (and low recombination).

I wasn't a reviewer during the first round of evaluation, but the authors also added mathematical models to their simulation outputs, following Abu Awad & Roze (2018) and Hayward and Sella (2022) theoretical frameworks. They also took into account previous reviewers' comments.

Overall, I found the manuscript to be very interesting, and generally well-written, even if sometimes hard to digest because of the several methods used. The outcomes of the manuscript are a mix of confirmatory results found in previous similar models (all of them being well developed in the introduction and discussion sections of the manuscript), with selfing being less disadvantageous than expected when considering a quantitative genetics framework, as already partially reviewed and summarize in Sztepanacz et al. (2023). However, the manuscript also brings novelties both in the analytical and theoretical modeling, notably for the mechanisms and signature of polygenic selection in predominantly selfing populations.

I think the manuscript is hence a good contribution to the field and is novel enough to be published Plos Genetics after a few modifications/clarifications.

Comments:

Material and method section: Some key information are missing for the simulation model to be fully replicated. Please indicate how selection and self-fertilization occur in the simulation, and if the mutation rate is kept constant both during the burn-in and selection periods, or if the mutation rate is changed with the environmental change.

L292 and equation 12b: Please either choose to note the inbreeding covariance C_I or V_I

L332: Please clarify if selection is possible thanks to the standing genetic variance accumulated at M-S-D or a mix of this diversity AND new mutations.

L374-392: Is it also possible that adaptation in highly selfing populations in your simulations results from both the contribution of standing diversity and the fixation (or increase in frequencies) of new mutations? Clo et al. (2020) only considered standing diversity (the mutation rate was fixed to 0 during directional selection). The contribution of both mechanisms makes patterns being different from that found Clo et al. 2020.

L720: Lande & Porcher 2015 used Kelly's work for sure, but the age of selfing paper is developed in Kelly (2007), not Kelly (1999) even if the concept is introduced within the structured linear model.

L820-822: I would refer to Manna et al. (2011) and add Martin et al. (2007) to introduce the notion of dominance and epistasis on the fitness scale.

References :

Kelly, J. K., 2007 Mutation-selection balance in mixed mating populations. J. Theor. Biol. 246: 355–365

Manna, F., Martin, G., & Lenormand, T. (2011). Fitness landscapes: an alternative theory for the dominance of mutation. Genetics, 189(3), 923-937.

Martin, G., Elena, S. F., & Lenormand, T. (2007). Distributions of epistasis in microbes fit predictions from a fitness landscape model. Nature genetics, 39(4), 555-560.

Sztepanacz, J., Clo, J., & Opedal, Ø. (2023). Evolvability, Sexual Selection, and Mating Strategies. Evolvability: A Unifying Concept in Evolutionary Biology?, 239-266.

Reviewer #3: The authors address the effects of self-fertilization on polygenic adaptation (namely, the population-genetic evolution of a quantitative trait under stabilizing selection) using a mathematical modeling approach complemented by computer simulations. It is well established that self-fertilization is widespread in natural populations. Therefore, a conceptual analysis of its effects on polygenic adaptation - as done in this work - is timely and relevant. In particular, it is interesting to better understand the interplay between the strength of direct selection on trait variants versus emerging genetic associations. Importantly, the authors manage to reasonably contrast their results against relevant, recent modeling papers that consider purely randomly mating populations, and related work in partially selfing individuals. In summary, the authors find that self-fertilization has opposing effects on the rate of adaptation and that its effects may even cancel under particular conditions.

Firstly, the authors set out by constructing a diploid two-locus model with potentially unequal effects to a quantitative trait. They study how allele frequencies and linkage disequilibrium (LD) change as one modulates the rates of recombination and self-fertilization, and characterize the effects of self-fertilization on LD at the old optimum (where it enhances LD), after the shift in optimum (where it has opposing effects on the rate of adaptation) and on the long term.

Secondly, the authors move to the n-locus case by expanding the modeling framework of Hayward and Sella (2022) by including genetic associations between loci and the rate of selfing as a parameter. With self-fertilization, they find that at the old optimum (i.e., mutation-selection balance) reduced purging increases the genetic variance. Neglecting genetic associations after the shift in optimum, the rapid phase remains unaltered (the increase in homozygosity and drift cancel under the employed conditions) while the later equilibration phase is sped up as the efficacy of selection is increased disproportionately w.r.t. drift. The situation is significantly more involved when genetic associations are accounted for and therefore, the authors employ computer simulations.

Thirdly, the authors employ individual-based computer simulations of polygenic adaptation (including neutral regions, background selection, and an expansion from single to multiple traits to include pleiotropy) to characterize the selection response under more realistic conditions. They find qualitative agreement with their analytical approximations, and rather close agreement with related work. Their simulations showcase that inbreeding yields major changes to the haplotype structure. However, under the studied conditions selfing does not strongly affect the rate of adaptation, and interestingly, highly selfing populations achieve a lower degree of inbreeding depression on the long term.

To conclude, this work draws our attention to the advantages of self-fertilization, contrary to the somewhat traditional belief that it hinders adaptation to new environments. Furthermore, this work provides major implications for the improvement of selection scans. Accordingly, this work will likely stimulate further advances in the near future.

The authors provide well annotated and detailed derivations in a set of supplementary files.

Minor remarks:

1. There are unclear formulations in the manuscript, e.g., on line 164 where the x_i and g_ij actually denote relative frequencies.

2. Furthermore, the paper requires another round of spell checking, e.g., see line 163.

**Have all data underlying the figures and results presented in the manuscript been provided?**

Reviewer #1: None

Reviewer #2: Yes

Reviewer #3: Yes

PLOS authors have the option to publish the peer review history of their article (what does this mean?). If published, this will include your full peer review and any attached files.

Reviewer #1: No

Reviewer #2: No

Reviewer #3: No

**Data Deposition**

http://datadryad.org/submit?journalID=pgenetics&manu=PGENETICS-D-24-00336

**Press Queries**

---

## [Editor Report · Acceptance letter]

12 Jul 2024

PGENETICS-D-24-00336 

Polygenic selection to a changing optimum under self-fertilisation 

Dear Dr Hartfield, 

We are pleased to inform you that your manuscript entitled "Polygenic selection to a changing optimum under self-fertilisation" has been formally accepted for publication in PLOS Genetics! Your manuscript is now with our production department and you will be notified of the publication date in due course.

With kind regards,

Zsofia Freund

PLOS Genetics

On behalf of:
